# Modafinil for Promoting Wakefulness in Critically Ill Patients: Current Evidence and Perspectives

**DOI:** 10.3390/clockssleep7040062

**Published:** 2025-10-27

**Authors:** Sotirios Kakavas, Dimitrios Karayiannis

**Affiliations:** 1Intensive Care Unit, Henry Dunant Hospital Center, 11526 Athens, Greece; 2Department of Clinical Nutrition, Evaggelismos General Hospital, 10676 Athens, Greece; jimkar_d@yahoo.com

**Keywords:** modafinil, critically ill patients, ICU, excessive daytime sleepiness, sleep disorders

## Abstract

Critically ill patients are predisposed to developing cognitive dysfunction, excessive daytime sleepiness (EDS), and fatigue during their stay in the intensive care unit (ICU). Modafinil, a wakefulness-promoting agent, has demonstrated potential benefits in enhancing alertness, cognitive performance, and activity levels in various clinical populations. The present narrative review aims to systematically evaluate the existing literature regarding the administration of modafinil for the treatment of EDS and fatigue in the ICU context. A comprehensive literature search was performed using the Embase, MEDLINE, Web of Science, and Google Scholar databases, covering publications up to 20 June 2025. Studies investigating the use of modafinil to improve wakefulness in ICU patients were identified. A total of nine relevant studies were included, comprising two randomized controlled trials (RCTs), two case series, and five retrospective cohort studies (*n* = 950 patients). Four of these studies focused on patients with traumatic brain injury or post-stroke conditions, whereas the remaining studies addressed heterogeneous ICU populations. Preliminary evidence indicates that modafinil may enhance wakefulness in selected critically ill patients and potentially facilitate their participation in rehabilitative interventions, such as physical therapy. Nonetheless, robust conclusions regarding efficacy and safety remain limited by the small sample sizes and methodological constraints of the available studies. Consequently, further large-scale RCTs are warranted to elucidate the therapeutic role of modafinil in the management of EDS and hypoactivity among ICU patients.

## 1. Introduction

It is well established that critical illness and ICU admission exert a profound impact on sleep regulation [1]. Excessive daytime sleepiness (EDS) and fatigue constitute pervasive and multifactorial problems that are frequently encountered in the intensive care unit (ICU) [2]. Within this clinical setting, the pathogenesis of EDS is heterogeneous and may involve: (i) inadequate sleep quantity, or sleep fragmentation due to underlying pathology, critical illness, pain, or environmental disturbances such as light exposure and noise; (ii) sleep-disordered breathing, particularly obstructive sleep apnea (OSA), which may be precipitated, exacerbated, or newly diagnosed during critical illness or recovery; (iii) circadian rhythm desynchronization as a result of the ICU environment; (iv) anesthetic and opioid administration, both of which increase the risk of OSA and excessive somnolence; and (v) neurological conditions, including traumatic and non-traumatic brain injury (TBI), or psychiatric complications, such as major depressive disorder developing during or after intensive care [1,2].

Modafinil, a non-amphetamine oral psychostimulant, is an agent that is uniquely approved by the U.S. Food and Drug Administration for the treatment of EDS associated with narcolepsy, residual EDS in OSA despite optimal continuous positive airway pressure (CPAP) therapy, and excessive sleepiness related to shift-work sleep disorder [3,4]. Although the precise mechanisms underlying its wakefulness-promoting effects remain incompletely elucidated, prior research suggests that modafinil possesses not only alertness-enhancing properties but also neuroprotective and antioxidative actions [3]. These pharmacological characteristics position modafinil as a potentially valuable therapeutic option for addressing EDS in the ICU. Accordingly, the present narrative review aims to synthesize and critically appraise the available literature on the administration of modafinil for the management of EDS and fatigue in critically ill patients.

## 2. EDS in Critically Ill Patients

Assessing sleepiness and fatigue in critically ill patients remains challenging. In ambulatory settings, validated self-report tools such as the Epworth Sleepiness Scale, the Stanford Sleepiness Scale, and the Fatigue Severity Scale are widely applied; however, these require intact cognition and communication, which are often impaired in the ICU by sedation, delirium, or encephalopathy. Consequently, clinicians rely on behavioral assessments. The Glasgow Coma Scale (GCS) provides a global measure of consciousness but distinguishes sleepiness from sedation, stupor, or deliriumpoorly. The Richmond Agitation–Sedation Scale (RASS) and Sedation–Agitation Scale (SAS) quantify agitation and sedation depth without directly measuring sleepiness, while the CAM-ICU and ICDSC detect delirium but are confounded by sedation. Objective techniques, including polysomnography, EEG, and actigraphy, offer insights into sleep–wake patterns, but face feasibility and accuracy limitations. No dedicated, validated tool currently exists for evaluating sleepiness or fatigue in critical illness, highlighting the urgent need for standardized ICU-specific instruments [5].

EDS can be defined as the inability to maintain wakefulness and alertness during the day, leading to unintended lapses into sleep. EDS is frequently observed in critically ill patients and represents a significant barrier to recovery in the ICU. In critically ill populations, EDS is multifactorial in origin, resulting from a complex interplay of disrupted sleep patterns, sedative medication effects, systemic inflammation, neurological dysfunction, and environmental factors [6]. Beyond the established contributors to excessive daytime sleepiness in critically ill patients, nonconvulsive status epilepticus (NCSE) should also be considered in the differential diagnosis. NCSE is increasingly recognized in the ICU, where it may manifest as otherwise unexplained somnolence, fluctuating levels of consciousness, or states mimicking a coma (EDS-coma). Prompt identification and treatment are critical, as delayed recognition is associated with increased morbidity, prolonged ICU stay, and adverse neurological outcomes [7]. Consequently, NCSE should be systematically included among the potential causes of “sleepiness” in the ICU. Importantly, EDS in ICU patients is not only a symptom but may also serve as a marker of sleep deprivation, delirium, or underlying neurological dysfunction. Its presence can delay mobilization, affect participation in physical therapy, and prolong ICU and hospital stays [6]. Figure 1 summarizes the contributory factors of excessive daytime sleepiness in the ICU.

Given these challenges, there is growing interest in interventions to improve sleep and reduce EDS. Identifying and managing EDS may play a vital role in improving both short- and long-term outcomes. Comprehensive management of EDS requires a multifaceted approach, including pharmacological and non-pharmacological strategies. Non-pharmacological measures, such as optimizing sedation protocols and promoting sleep hygiene with noise reduction and light therapy should be considered [8]. Pharmacologic approaches, including agents like modafinil, have been explored, but require further investigation to establish their safety and efficacy in the critically ill [1,2].

### 2.1. Sleep Disruption and Circadian Desynchronization in the ICU

Sleep disruption arising from environmental disturbances is a primary contributor to EDS in the ICU. Continuous lighting, frequent nursing interventions, alarms, and mechanical ventilation interfere with both sleep initiation and maintenance. The levels of noise in the ICU environment commonly exceed recommended thresholds [9]. Poor sleep efficiency and prolonged sleep onset (sleep latency) are commonly present. These brief sleep periods of ICU patients are interrupted by frequent arousals and are evenly distributed over the day and night [1,2]. Sleep architecture is significantly altered. Sleep in critically ill patients is characterized by reductions in restorative slow-wave and REM sleep, while being dominated by lighter stages (N1 and N2) [10]. These alterations contribute to prolonged fatigue and impaired neurocognitive recovery, and they often persist beyond ICU discharge [1,2].Moreover, the acute stress response and systemic inflammation associated with critical illness may further disrupt sleep–wake regulation, possibly through the dysregulation of cytokines like interleukin-6 and tumor necrosis factor-alpha, which are known to affect sleep architecture [11]. Finally, constant exposure to artificial light and lack of temporal cues may lead to the loss of circadian rhythm and exacerbate sleep fragmentation. In critical care settings, desynchronization of melatonin secretion and disruption of the circadian cycle are common, impacting patients’ health and recovery [12,13].

### 2.2. OSA in the ICU

Obstructive sleep apnea (OSA) represents one of the most frequent causes of excessive daytime sleepiness in the general population, and is increasingly recognized among critically ill patients. Many ICU patients are obese or have pre-existing risk factors that predispose them to upper airway collapse during sleep. However, in the context of endotracheal intubation, the artificial airway bypasses the upper airway and effectively eliminates OSA events. For this reason, OSA is unlikely to be a relevant factor in excessive sleepiness during the intubated phase of critical illness. The situation changes substantially during weaning and post-extubation recovery. As patients transition from mechanical ventilation, a number of factors can reintroduce or exacerbate sleep-disordered breathing: residual sedatives and opioids reduce airway tone and blunt arousal responses; obesity and fluid shifts increase peripharyngeal edema; and prolonged intubation may cause laryngeal trauma, edema, or vocal cord dysfunction, further narrowing the airway [14,15,16,17,18]. Supine positioning and ICU-acquired neuromuscular weakness may aggravate these risks, making previously mild or subclinical OSA clinically significant after extubation.

Clinically, this means that OSA should not be overlooked when ICU patients demonstrate unexplained hypersomnolence, failed weaning attempts, or persistent fatigue in the days following extubation. Indeed, sleep-disordered breathing has been described as an under-recognized contributor to prolonged ventilator dependence and impaired recovery [16]. Recognizing the temporal dynamics of OSA in the ICU—largely irrelevant during intubation but potentially important during liberation from mechanical ventilation—can guide both diagnosis and management. Addressing OSA in this vulnerable period may not only improve sleep quality and alertness but also facilitate rehabilitation and shorten ICU stays.

### 2.3. Critical Illness and Underlying Disease

EDS represents a common and often persistent complication following acute brain injury in critically ill patients, including those with traumatic brain injury (TBI), stroke, subarachnoid hemorrhage, or hypoxic–ischemic encephalopathy. In these patients, the pathophysiology of EDS is multifactorial, but injury to key brain regions involved in sleep–wake regulation plays a central role [19,20]. However, even in critically ill patients without acute brain injury, systemic inflammation and critical illness-related brain dysfunction also contribute to EDS [1,2]. In conditions such as sepsis, delirium, and encephalopathy, brain regions responsible for wakefulness can be affected, further promoting hypersomnolence. For example, elevated levels of proinflammatory cytokines such as interleukin-1 (IL-1), interleukin-6 (IL-6), and tumor necrosis factor-alpha (TNF-α) may disrupt neurotransmission and affect the hypothalamic centers that control arousal and sleep [10]. Likewise, impaired circadian melatonin secretion in septic patients is mainly related to the presence of severe sepsis and/or concomitant medication [21]. Importantly, inflammation and critical illness-related neuromuscular dysfunction may compromise the central regulation of breathing and further impair the tone of the pharyngeal dilator muscles, particularly during sleep [22]. ICU-acquired weakness and prolonged immobility also play significant indirect roles, leading to physical deconditioning and fatigue [22].

### 2.4. Medication and EDS in the ICU

Pharmacological agents used in ICU management also play a significant role in EDS, by disrupting normal sleep architecture or exerting residual sedative effects that impair daytime alertness [1,2]. Most drugs with clinical sedative or hypnotic actions affect one or more of the central neurotransmitters implicated in the neuromodulation of sleep and wakefulness. Yet, numerous non-sedative drugs can impair normal sleep architecture [23]. Polypharmacy, drug accumulation due to organ dysfunction, and prolonged drug half-lives in critically ill patients further heighten these risks [23]. Sedative-hypnotics, such as benzodiazepines and propofol, are significant contributors to EDS. These drugs can reduce slow-wave and REM sleep—the most restorative stages—resulting in non-restorative sleep and persistent somnolence [8]. While sedation induces a non-physiological sleep-like state, this is often restorative rest, leading to residual sleepiness during waking hours [9]. Additionally, prolonged sedation may impair the reestablishment of normal wakefulness and circadian rhythm once sedation is weaned. Similarly, opioids administered for pain control also interfere with sleep continuity and decrease REM sleep [23]. Moreover, these drugs also depress the respiratory drive and may exacerbate sleep-disordered breathing, leading to fragmented sleep patterns [24]. Finally, antipsychotics, often used to manage ICU delirium, can induce sedation and worsen EDS, particularly those with strong antihistaminergic effects, like quetiapine [23].

## 3. Pharmacological Properties of Modafinil

### 3.1. Mechanism of Action

Modafinil and its R entaniomer, armodafinil, are nonamphetamine wakefulness-promoting medications that are considered to be the first-line medication for the treatment of EDS associated with narcolepsy [3]. Residual EDS in OSA and excessive sleepiness associated with shift-work sleep disorder have also been shown to respond to modafinil [4]. Modafinil enhances function in a number of cognitive domains and has also been used, off label, as a cognitive enhancer in healthy populations or patients with impaired cognitive function. The mechanism of action for this agent is still unclear [4]. Most likely, modafinil acts by enhancing nor-epinephrine (NE) and the dopaminergic transmission of many brain sites associated with wakefulness [3,4]. Previous research showed that modafinil promotes the activity of various wake-promoting centers, including the tuberomammillary nucleus (TMN) and hypocretin cells of the perifornical area [25]. These centers receive dopaminergic innervations, and modafinil may increase dopaminergic signaling by altering the dopamine reuptake [26,27]. Modafinil binds weakly to the dopamine transporter (DAT), and it has been reported that the drug does not improve wakefulness in mice lacking the DAT [26]. Furthermore, modafinil may inhibit the activity of the ventrolateral preoptic nucleus (VLPO) by blocking the reuptake of NE bynoradrenergic neural terminals in the VLPO [27]. The mechanism of action for modafinil is distinct from classical psychostimulants, in terms of the involvement of the histaminergic and the orexinergic systems. Modafinil indirectly activates the histaminergic system, presumably via the attenuation of the inhibitory GABAergic input to the histaminergic neurons [3,28]. Moreover, evidence shows that modafinil increases histaminergic tone via orexinergic neurons [3]. Finally, it has been reported that modafinil possesses anti-oxidative properties, and it could protect the striatum against protein oxidative damage and hinder the sleep-promoting effects of free radicals [3]. Figure 2 schematically depicts the major neurochemical pathways affected by modafinil.

### 3.2. Pharmacokinetics and Dosing

Modafinil is a racemic mixture containing both L and R enantiomers, with half-lives of 3–4 hand 10–14 h, respectively [4,29]. The absorption of the drug is quite fast and peak plasma concentrations occur at 2–4 h after administration. Usually, modafinil is administered orally, once daily, in the morning (200–400 mg). The maximum recommended daily dose of modafinil is 400 mg [30]. However, higher doses may be required for the adequate control of sleepiness [31]. Since the elimination half-life of modafinil is 9 to 14 h, once-daily administration is sufficient for most patients. Yet, in a sub-group of patients, poor control of daytime sleepiness may be noticed in the afternoon or early evening. In this case, a split dosing (200 mg in AM, 200 mg at 1–2 PM) may be effective [30,31].

### 3.3. Guiding Principles for Commencing and Ceasing Modafinil Treatment

While pharmacokinetics guide dosing, the decision to initiate or discontinue modafinil is equally critical, particularly in the setting of critical illness. Initiation should be reserved for patients in whom excessive daytime sleepiness (EDS) persists, despite optimized management of underlying conditions, including obstructive sleep apnea, narcolepsy, or postoperative sedation, and in whom sleepiness impairs participation in rehabilitation, cognitive engagement, or recovery activities [4]. Prior to therapy, secondary contributors to somnolence—such as sedative medications, metabolic derangements, delirium, or circadian disruption—should be carefully addressed. In critically ill or postoperative patients, modafinil should not be administered during the acute unstable phase. The optimal window for initiation is typically post-acute stabilization, approximately 48–72 h after ICU admission or major surgery, once sedatives and analgesics have been minimized and neurological responsiveness allows for meaningful participation in mobilization or cognitive assessments [6]. In neurocritical care populations, such as post-stroke or traumatic brain injury, initiation may be considered as early as days 3–5, when cognitive function can be reliably assessed. Therapy should be goal-directed, supporting functional recovery rather than serving as routine pharmacological maintenance.

Discontinuation should be considered when EDS resolves; functional wakefulness is restored; or adverse effects such as agitation, insomnia, or hypertension occur. Modafinil’s low potential for physiological dependence typically obviates the need for tapering; however, structured monitoring for the recurrence of EDS is advised. Premature initiation in postoperative settings may interfere with analgesic regimens or exacerbate anxiety, and ongoing delirium remains a contraindication until resolution. In the absence of high-level randomized controlled trial data, the timing of initiation and discontinuation must rely on individualized clinical judgment, guided by stabilization, functional recovery, and ongoing reassessment of the risk–benefit balance.

### 3.4. Side Effects and Drug Interactions

Headaches and nausea are the most common side-effects of modafinil. Headaches can be managed with simple analgesics or minimized by a slow increase in dose [4,32]. Other mild side-effects that have been described include dizziness, diarrhea, dry mouth, nose and throat congestion, back pain, insomnia, and mental side effects like anxiety and nervousness [32]. In rare cases, modafinil can cause serious side effects such as a severe skin rash (Stevens–Johnson syndrome) or psychiatric symptoms like hallucinations and mania. The evidence shows that modafinil exhibits no or limited tolerance, and when discontinued, a rebound of REM and slow wave sleep is unlikely [33]. The metabolism of the drug depends mainly on the hepatic cytochrome P-450 (CYP450) system [4]. Moreover, modafinil is also potent to suppress various numbers of cytochrome isoenzymes. Thus, a number of drug interactions may occur. For example, co-administration of modafinil may increase the circulating levels of diazepam, warfarin, phenytoin, and propranolol [4,32]. In some cases, an adjustment of the drugs’ dosages may be required.

## 4. Administration of Modafinil in Critically Ill Patients

As previously discussed, the clinical course of patients admitted to the intensive care unit (ICU) is frequently complicated by the development of excessive daytime sleepiness (EDS) and fatigue. These conditions arise through a multifactorial interplay of environmental disturbances, the underlying critical illness, neuropsychiatric complications, and the adverse effects of pharmacological interventions, collectively contributing to cognitive impairment, neuromuscular dysfunction, sleep deprivation, sleep fragmentation, and sleep-disordered breathing [1,2].

Within this setting, modafinil exhibits a particularly favorable pharmacological profile as a wakefulness-promoting agent. Its advantages include convenient oral administration, documented efficacy and safety across a range of patient populations with EDS, and a low potential for abuse, dependence, or severe drug–drug interactions [3,4]. In contrast to classical psychostimulants such as amphetamines, modafinil does not appear to significantly disrupt nocturnal sleep architecture, and is not typically associated with marked behavioral excitation or rebound hypersomnolence [34,35].

Evidence from non-ICU populations further supports its therapeutic potential. Modafinil has been shown to alleviate EDS and fatigue in patients with amyotrophic lateral sclerosis [36], multiple sclerosis [37], cancer-related or opioid-induced fatigue [38,39], and major depressive disorder [40,41]. Additionally, beneficial effects have been reported in perioperative settings, where modafinil improved patient-reported fatigue and enhanced postoperative recovery [42,43]. Given that obstructive sleep apnea (OSA) is prevalent among ICU patients, particularly in the perioperative context, it is notable that modafinil has demonstrated efficacy in improving residual EDS, functional performance, and quality of life in patients, with OSA receiving optimal continuous positive airway pressure (CPAP) therapy [2,44].

Nevertheless, despite these promising findings, evidence directly addressing the role of modafinil in critically ill populations remains limited. To date, only a small number of randomized controlled trials (RCTs) and observational studies have investigated its efficacy in the ICU context. Accordingly, for the present review, a systematic literature search was undertaken to identify studies evaluating the use of modafinil in critically ill patients, with a primary focus on wakefulness promotion in the ICU. Searches were conducted in Embase, MEDLINE, Web of Science, and Google Scholar for studies published from January 2008 through to 31 July 2025 in the English language. The reference lists of relevant articles were also screened to identify additional eligible studies. Inclusion was restricted to studies reporting on the administration of modafinil in critically ill ICU populations, while investigations of modafinil in other conditions (e.g., narcolepsy, shift work disorder, or non-ICU cohorts) were excluded. The search strategy was constructed using a combination of Medical Subject Headings (MeSH), Emtree terms, and free-text keywords. Boolean operators were applied to ensure both sensitivity and specificity of the retrieved records. The core concepts included the following. Intervention: “modafinil”, Population/Setting: “critically ill”, “critical illness”, “intensive care unit”, “ICU”, andOutcomes: “excessive daytime sleepiness”, “EDS”, “fatigue”, “sleep disorders”, and “hypersomnolence”.

In total, 10 studies met the eligibility criteria, comprising three randomized controlled trials, two case series, and five retrospective cohort studies (*n* = 950 patients). Of these, four specifically investigated patients with traumatic brain injury (TBI) or post-stroke conditions, whereas the remaining studies encompassed heterogeneous ICU populations. A detailed summary of the included studies is provided in Table 1.

### 4.1. Mixed ICU Populations

In 2015, Gajewski et al. reported a case series of three critically ill patients who received modafinil during their stay in a Thoracic Surgery ICU [45]. The patients’ selection was based on the hypothesis that their clinical course was complicated by the presence of fatigue, EDS, and/or depression, and that modafinil administration could promote the patients’ wakefulness and participation. Hence, 200 mg of modafinil were administered orally each morning, and according to the authors, the patients responded with a marked improvement of alertness, activeness, and participation in physical therapy during the following days. An improved sleep regulation was also noticed. In a second, more recent, case series with eight critically ill patients (including COVID-19 patients) requiring mechanical ventilation, modafinil was administered to promote ICU wakefulness. Modafinil in a daily dose of 100–200 mg for a median duration of four days led to a GCS improvement in five (62.5%) patients. Furthermore, modafinil prevented tracheostomy in one COVID-19 patient. No significant adverse effects were documented [46].

A retrospective cohort study by Mo et al. studied the effect of modafinil on the alertness cognitive function of critically ill patients [47]. The study included 60 ICU patients requiring ventilatory support, either invasive or noninvasive, at the time of modafinil initiation. The patients received an average daily modafinil dose of 170 mg for a median duration of nine days, and the average scores of the Glasgow Coma Scale (GCS) and Riker Sedation-Agitation Scale (SAS) were recorded for 48 h before and after the start of modafinil therapy. According to the results, after controlling for possible confounding factors, modafinil treatment was associated with a small, but non-significant increase in average scores of GCS by 0.34 points. Most importantly no modafinil-related side effects were detected. Modafinil administration was not associated with a significant change in the average SAS scores. The administration of modafinil in ICU patients appears to be safe, and most of the studies report a low frequency of adverse events. Nevertheless, a retrospective chart study [48] reported that critically ill patients who received modafinil were characterized by an increased duration of delirium, the length of their hospital stay, and mechanical ventilation, compared with their matched controls. On the other hand, another retrospective chart study [49] in ICU patients intubated for at least five days or more showed a non-significant reduction in the mean number of days to wean from ventilation with modafinil. Likewise, an RCT by Mansouri et al. examined the effect of modafinil administration on cognitive dysfunction, weaning, and hospital stay after on-pump coronary artery bypass grafting surgery [50]. The patients received 200 mg of modafinil on the day of the surgery, and 200 mg the morning after surgery. The results showed that modafinil was associated with significantly decreased time to reach consciousness, ventilator time in the ICU, length of stay in ICU, duration of hospitalization, and arterial blood carbon dioxide pressure. No major adverse effects were noted. These results are in accordance with a previous prospective, randomized, double-blind study, reporting that modafinil in a single postoperative dose of 200 mg significantly improved self-reported recovery variables after general anesthesia, including alertness and fatigue [43].

### 4.2. ICU Patients with Traumatic and Non-Traumatic Brain Injury

The evidence concerning the potential benefits of modafinil in the critically ill population seems to be more compelling in patients with traumatic brain injury (TBI). In an earlier RCT including patients with chronic TBI, modafinil ameliorated EDS but not fatigue at week 4, but not at week 10, compared with a placebo [55]. Modafinil was safe and well tolerated, although insomnia was reported significantly more often with modafinil. In an RCT conducted by Kaiser et al., the administration of 100 to 200 mg of modafinil in 20 ICU patients with TBI who had fatigue, EDS, or both, was well tolerated and ameliorated by EDS [51]. Although their fatigue was not improved, the performance during the maintenance of wakefulness test improved in the modafinil group. Likewise, a retrospective study in ICU patients treated with amantadine and/or modafinil following TBI showed that the median GCS increased by a median of one, and that the treatment weas well tolerated [52]. More recently, Zand et al. published the results of double-blind RCT about the efficacy of oral modafinil on the enhancement of consciousness recovery in adult ICU patients with moderate to severe acute TBI [53]. The analysis showed a significantly greater proportion of patients with an increase in total GCS by one or two units in the modafinil group (56% vs. 34% and 54% vs. 32%, respectively). On the other hand, mixed results concerning the effectiveness of modafinil on wakefulness in the ICU were provided by Leclerc et al. [54]. This retrospective study evaluated 87 ICU patients treated with amantadine and/or modafinil following acute non-traumatic intracerebral hemorrhage (ICH), ischemic stroke (IS), or subarachnoid hemorrhage (SAH). Indications for the administration of amantadine and/or modafinil included somnolence (77%), not following commands (32%), lack of eye opening (28%), or low GCS (17%). The most common initial dose was 100 mg twice daily for both drugs. According to the results, 55% of the patients receiving amantadine monotherapy and 33% of them receiving both amantadine and modafinil were considered responders. The rate of discharge was significantly higher in the responders group. However, no responders were noticed among the 15 patients that received modafinil monotherapy. Yet, it should be noted that modafinil was the initial neurostimulant administered in only 15% of the patients. Modafinil was discontinued in only one patient, due to insomnia and agitation. A series of other studies in mixed populations of patients with stroke reported a beneficial effect of modafinil on various outcomes [36,37,38,39], including EDS and fatigue [56,57,58,59].

## 5. Conclusions

Daytime sleepiness and fatigue in critically ill patients represent complex and multifactorial syndromes, influenced by sedation practices, pharmacologic therapies, muscle weakness, underlying illness, and psychological factors. The emerging, albeit limited, literature suggests that modafinil, administered orally, may promote wakefulness in selected patients within the intensive care setting. The available randomized controlled trials and case reports, although few in number, indicate that modafinil is generally safe and may be particularly valuable in hypoactive patients whose impaired arousal limits participation in mobilization or hinders liberation from mechanical ventilation. From a clinical perspective, modafinil should not be regarded as a substitute for established approaches to sedation minimization, delirium prevention, and rehabilitation in the ICU, but rather as a potential adjunctive agent when excessive daytime sleepiness persists despite the optimization of standard care. Its ability to enhance wakefulness could facilitate earlier engagement in physical and cognitive therapy, thereby supporting recovery trajectories and possibly reducing the length of ICU stays. Nevertheless, the existing evidence remains insufficient to define the populations most likely to benefit from it or the long-term implications of its use in critical illness with certainty. Future investigations should move beyond demonstration of feasibility and instead delineate the clinical scenarios in which modafinil provides measurable advantages. Such studies must clarify not only its effects on short-term outcomes, such as wakefulness and ventilator weaning, but also on longer-term cognitive and functional recovery after ICU discharge. In this way, the therapeutic role of modafinil in the management of hypoactivity and excessive daytime sleepiness in the ICU can be more precisely established.

## Figures and Tables

**Figure 1 clockssleep-07-00062-f001:**
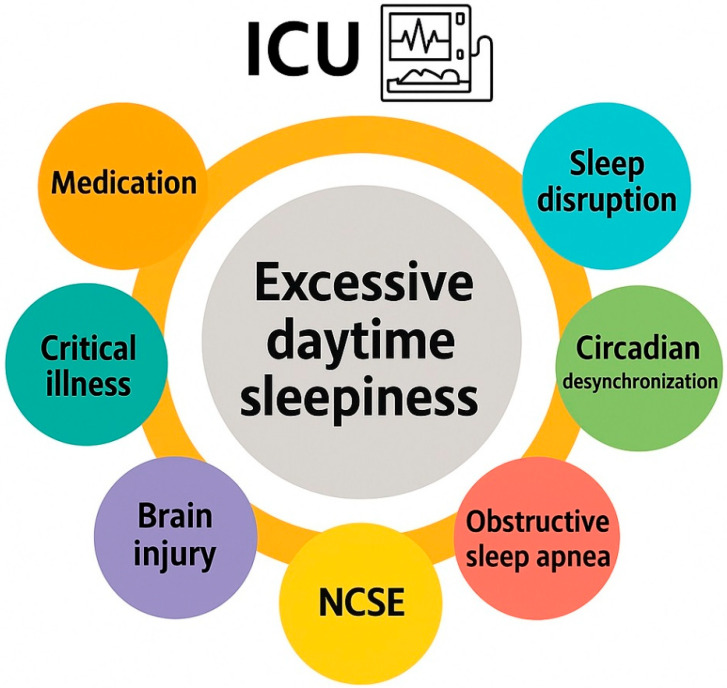
Contributory factors of excessive daytime sleepiness in the intensive care unit.

**Figure 2 clockssleep-07-00062-f002:**
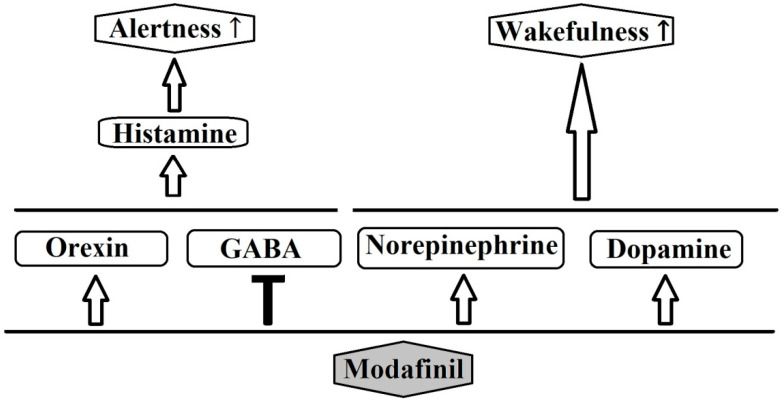
Possible pathways of the pharmacological action of modafinil.

**Table 1 clockssleep-07-00062-t001:** Studies on modafinil use in critically ill patients during intensive care unit stay.

Author (Ref).	Country (Year)	Study Type	Patient Population (*n* = Number of Patients)	Modafinil Daily Dosage	Outcome
Gajewski and Weinhouse, [45]	USA (2015)	Case series	Thoracic surgery ICU (*n* = 3)	100–200 mg	Improved alertness and activeness observed
Amer et al. [46]	Saudi Arabia (2022)	Case series	Mixed ICU population (*n* = 8)	100–200 mg	GCS improvement in 5 patients
Mo et al. [47]	USA (2017)	Retrospective	Mixed ICU population with ventilatory support (*n* = 60)	Median dose 170 mg	Non-significant increase in GCS by 0.34 points
Branstetter et al. [48]	USA (2022)	Retrospective	Mixed ICU population—CAM-ICU positive patients (*n* = 54)	Not mentioned	Increased duration of delirium, LOS and mechanical ventilation in the modafinil group
Keeney CP and Adeniyi [49]	USA (2022)	Retrospective	Mixed ICU population on mechanical ventilation (*n* = 511)	Not mentioned	Non-significant reduction in mean number of days to wean from ventilation with modafinil
Mansouri et al. [50]	Iran (2021)	RCT	Patients undergoing on pump CABG (*n* = 74)	200 mg on the day of surgery, and 200 mg the morning after surgery	Significant decreases in time to consciousness, ventilator time in ICU LOS in the ICU, duration of hospitalization and PaCO_2_
Kaiser et al. [51]	Switzerland (2010)	RCT	TBI (*n* = 20)	100–200 mg for 6 weeks	EDS improvement, no effect on fatigue
Barra et al. [52]	USA (2020)	Retrospective	TBI (*n* = 48)	100–150 mg	Median GCS increase by a median of 1, well tolerated
Zand et al. [53]	Iran (2024)	RCT	TBI (*n* = 85)	200 mg	Higher proportion of patients with an increase in total GCS in the modafinil group
Leclerc et al. [54]	USA (2021)	Retrospective	ICU patients with nontraumatic ICH, IS, or SAH (*n* = 87)	100 mg twice daily	Response 33% (amantadine and modafinil cotreatment) and 0% (modafinil monotherapy)

GCS: Glasgow Coma Scale, ICH: intracranial bleeding, IS: ischemic stroke, ICU: intensive care unit, RCT: randomized controlled trial, TBI: traumatic brain injury, SAH: subarachnoid hemorrhage, CAM-ICU: confusion assessment method for the ICU, CABG: coronary artery bypass grafting, LOS: length of stay, and PaCO_2_: arterial blood carbon dioxide pressure.

## Data Availability

No new data were created or analyzed in this study.

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
