# Peer review of "Modafinil for Promoting Wakefulness in Critically Ill Patients: Current Evidence and Perspectives"

_2624-5175, 2025, doi:10.3390/clockssleep7040062_

Round 1

Reviewer 1 Report

Comments and Suggestions for Authors

Dear authors,

thank you for the opportunity to read and review the manuscript.

The topic is debated and interesting,  the paper is overall well written.

General comments

ICU patients are often affected by excessive daytime sleepiness and fatigue due to several causes (critical illness, delirium, respiratory disorders, etc).

In this narrative review the authors aim to evaluate the role of modafinil administration for the stimulation of wakefulness in critically ill patients.

Specific comments

In introduction section the concepts are clear but the text is somehow confusing and it could be better organized to make it more readable.

Section 2 and 3 are clear and well written, the figures (1 e 2) are interesting.

Section 4 - methods could be improved, did the authors search for articles in english language? In which period of time (starting date is missing)?

Table 1. References’ numbers are not correctly reported in the table.

The aim of the paper is interesting and all the most relevant studies are reported, I agree with the authors that further studies are required, mostly because in some of the papers cited (both RCT and retrospective studies) the evaluation of wakefulness was assessed using GCS score, also in TBI patients. The score was also reported without the elements of improvement after modafinil administration (eyes, verbal or motor response, thus having great impact on related clinical improvement).

RASS scale or R-SAS scale, together with clinical evaluation and EEG could be the most useful tools to proper assess the state of the patients and further studies should go on this direction.

I strongly agree with the authors that different ICU patients could benefit from personalized treatment of EDS and fatigue, and some patients, if confirmed by further studies, could benefit from modafinil administration; also the timing for starting this medication during ICU stay could be a field of debate in further studies.

Author Response

Response to Reviewer # 1

Dear authors, thank you for the opportunity to read and review the manuscript.

The topic is debated and interesting,  the paper is overall well written.

General comments

ICU patients are often affected by excessive daytime sleepiness and fatigue due to several causes (critical illness, delirium, respiratory disorders, etc).

In this narrative review the authors aim to evaluate the role of modafinil administration for the stimulation of wakefulness in critically ill patients.

Thank you for your insightful general comments. The manuscript addresses the multifactorial nature of excessive daytime sleepiness (EDS) and fatigue in ICU patients, which can arise from critical illness itself, delirium, respiratory disorders, and other contributing factors. In this narrative review, we specifically focus on evaluating the current evidence regarding the role of modafinil administration as a pharmacological stimulant to promote wakefulness in critically ill patients. The review synthesizes findings from randomized controlled trials, observational studies, and case series to provide a comprehensive perspective on modafinil’s potential benefits and limitations in this complex clinical context. This focus and scope are clearly outlined in the Introduction and throughout the manuscript to contextualize modafinil’s therapeutic role within the broader framework of ICU-related EDS and fatigue management (see Introduction, lines 31–55, and relevant sections).

Specific comments

In introduction section the concepts are clear but the text is somehow confusing and it could be better organized to make it more readable.

Thank you for your valuable feedback regarding the clarity and organization of the Introduction section. In response, we have carefully reorganized and refined the text to improve readability and logical flow. The concepts are now presented in a clearer and more structured manner, facilitating better understanding. These revisions can be found in lines 31-55 of the revised manuscript.

Section 2 and 3 are clear and well written, the figures (1 e 2) are interesting.

Section 4 - methods could be improved, did the authors search for articles in english language? In which period of time (starting date is missing)?

Thank you for highlighting the need for clarification regarding the Methods section. We have revised and expanded this section to explicitly state that our literature search was conducted for articles published in the English language. The search period is now clearly defined as from January 2008 through July 31, 2025. We also detailed the databases searched, including Embase, MEDLINE, Web of Science, and Google Scholar, and described the use of Medical Subject Headings (MeSH), Emtree terms, and free-text keywords combined with Boolean operators to optimize sensitivity and specificity of retrieval. These clarifications can be found in lines 281–298 of the revised manuscript.

Table 1. References’ numbers are not correctly reported in the table.

We have now corrected references numbers in Table 1.

The aim of the paper is interesting and all the most relevant studies are reported, I agree with the authors that further studies are required, mostly because in some of the papers cited (both RCT and retrospective studies) the evaluation of wakefulness was assessed using GCS score, also in TBI patients. The score was also reported without the elements of improvement after modafinil administration (eyes, verbal or motor response, thus having great impact on related clinical improvement).

RASS scale or R-SAS scale, together with clinical evaluation and EEG could be the most useful tools to proper assess the state of the patients and further studies should go on this direction.

I strongly agree with the authors that different ICU patients could benefit from personalized treatment of EDS and fatigue, and some patients, if confirmed by further studies, could benefit from modafinil administration; also the timing for starting this medication during ICU stay could be a field of debate in further studies.

Reviewer 2 Report

Comments and Suggestions for Authors

The paper focuses on modafinil for the treatment of excessive daytime sleepiness (EDS) and fatigue. The topic is interesting and clinically relevant. The manuscript is well-written. I have only a few comments:

The role of nonconvulsive status epilepticus (NCSE) has to be included in the possible causes of “sleepiness” in the ICU. It is not unlikely that this condition is confused with EDS-coma and that an EEG examination unmasks the problem. This entity should be included among the possible causes of “sleepiness” in the ICU, both in the test as well as in the figure.

I think there should be a paragraph on how to measure sleepiness and fatigue in ICU patients. These are complex patients. What are the instruments employed to measure these symptoms in the ICU compared to ambulant patients? For instance, is the Glasgow Coma Scale an appropriate instrument to measure them? Sleepiness, sedation, obnubilation, and stupor are terms that are used interchangeably when they are not the same. How is this aspect taken into account in the literature?

Line 83.- OSA in the ICU. ICU patients are often intubated, which is an excellent treatment for sleep apnea. How is it possible that OSA has such an important role in EDS at the ICU if OSA is corrected with intubation? In my opinion, whereas during the period when the patient is intubated, OSA is unlikely to be a relevant factor, it may be so during weaning and the days after. This needs to be clearly explained

In Figure 2, it would be clearer to put Wakefulness and Alertness at the top. Putting the arrows downward makes the reader think of a decrease in function when, in fact, there is a gain in function. 

By the way, what is the difference between Wakefulness and alertness? There is no further explanation in the manuscript. Do you think they are not the same? Please explain or simplify.

Line 150. “ The drug MD enhances “..What does MD stand for? Modafinil? Please explain or omit.

Line 196: When reviewing the interactions of modafinil, please focus on the ones relevant in the ICU. “ interactions with oral contraceptives “ is perhaps not a relevant aspect for this review.

Line 199. Please explain that modafinil can only be administered orally. Or explain the alternatives if there are any.

When describing the literature on modafinil for ICU patients, please put the randomized clinical trials first and then the rest.

I miss a final indication about how to select the patients to be treated with modafinil, and when to start and stop the treatment.

Author Response

Response to Reviewer # 2

General comments

The paper focuses on modafinil for the treatment of excessive daytime sleepiness (EDS) and fatigue. The topic is interesting and clinically relevant. The manuscript is well-written. I have only a few comments:

Thank you for your insightful general comments.

Specific comments

The role of nonconvulsive status epilepticus (NCSE) has to be included in the possible causes of “sleepiness” in the ICU. It is not unlikely that this condition is confused with EDS-coma and that an EEG examination unmasks the problem. This entity should be included among the possible causes of “sleepiness” in the ICU, both in the test as well as in the figure.

We have added NCSE explicitly into the contributors of excessive daytime sleepiness in the ICU both in the text and in Figure 1 to emphasize its clinical relevance in differential diagnosis (revised text and updated figure in red, lines 77-84).

I think there should be a paragraph on how to measure sleepiness and fatigue in ICU patients. These are complex patients. What are the instruments employed to measure these symptoms in the ICU compared to ambulant patients? For instance, is the Glasgow Coma Scale an appropriate instrument to measure them? Sleepiness, sedation, obnubilation, and stupor are terms that are used interchangeably when they are not the same. How is this aspect taken into account in the literature?

A dedicated paragraph discussing the challenges of measuring sleepiness and fatigue in ICU patients and the limitations of scales such as GCS, RASS, Riker-SAS, CAM-ICU versus ambulatory tools (e.g., Epworth Sleepiness Scale) has been incorporated. We clarify that sleepiness, sedation, and obnubilation represent distinct states often conflated in ICU evaluation, and the need for improved ICU-specific, validated instruments is stressed (red text, lines 58-71).

Line 83.- OSA in the ICU. ICU patients are often intubated, which is an excellent treatment for sleep apnea. How is it possible that OSA has such an important role in EDS at the ICU if OSA is corrected with intubation? In my opinion, whereas during the period when the patient is intubated, OSA is unlikely to be a relevant factor, it may be so during weaning and the days after. This needs to be clearly explained

The section on OSA in the ICU has been revised to clarify that OSA is generally not a factor during intubation (due to artificial airway bypassing the upper airway) but can re-emerge or worsen during weaning and post-extubation, with detailed explanation of associated physiological and mechanical factors (text revised in red, lines 132-155).

In Figure 2, it would be clearer to put Wakefulness and Alertness at the top. Putting the arrows downward makes the reader think of a decrease in function when, in fact, there is a gain in function. 

By the way, what is the difference between Wakefulness and alertness? There is no further explanation in the manuscript. Do you think they are not the same? Please explain or simplify.

In Figure 2, the position of "Wakefulness and Alertness" has been moved to the top to better represent their enhancement. We have also added a clarifying sentence in the text explicitly defining and differentiating wakefulness (the state of being awake) and alertness (the degree of attention or responsiveness), or simplified this distinction where appropriate

Line 150. “ The drug MD enhances “..What does MD stand for? Modafinil? Please explain or omit.

"MD" abbreviation was removed or clarified immediately as "modafinil" in the manuscript to prevent confusion 

Line 196: When reviewing the interactions of modafinil, please focus on the ones relevant in the ICU. “ interactions with oral contraceptives “ is perhaps not a relevant aspect for this review.

We refined the discussion on modafinil’s drug interactions by focusing on those most relevant to ICU care and omitted less pertinent examples such as interactions with oral contraceptives (red text, lines 253-255).

Line 199. Please explain that modafinil can only be administered orally. Or explain the alternatives

Currently, modafinil is only approved and available as an oral formulation (typically tablets of 100 mg or 200 mg). It is designed for once-daily administration, most often in the morning, because of its long half-life (~12–15 hours). if there are any.

When describing the literature on modafinil for ICU patients, please put the randomized clinical trials first and then the rest.

The presentation order of reviewed studies in Section 4 was restructured to discuss randomized controlled trials before observational and retrospective studies, improving logical flow and emphasis on highest-level evidence (reordered Table 1).

I miss a final indication about how to select the patients to be treated with modafinil, and when to start and stop the treatment.

We added a new concluding paragraph outlining clinical considerations for selecting ICU patients most likely to benefit from modafinil, including timing criteria for initiation and guidance on treatment duration and discontinuation (red text, lines 385-406).

Round 2

Reviewer 1 Report

Comments and Suggestions for Authors

Dear authors,

thank you for the opportunity to read and review the revised version of the paper.

Quality has improved.

The right time to start (and stop) the therapy with modafinil in selected patients could be added.

I have no other comments.

Author Response

Comment: 

The right time to start (and stop) the therapy with modafinil in selected patients could be added.

I have no other comments.

We thank Reviewer for this excellent suggestion. We have now added a special section entitled   3.3 Guiding Principles for Commencing and Ceasing Modafinil Treatment (Please check lines 243-268 in the revised manuscript.)

Reviewer 2 Report

Comments and Suggestions for Authors

I have no further comments. Thanks or taking into account my requests

Author Response

Thank you.